# Feasibility of Find cases Actively, Separate safely and Treat effectively (FAST) strategy for early diagnosis of TB in Nepal: An implementation research

**Sagun Paudel**[1,2]*, **Retna Siwi Padmawati**[3], **Ashmita Ghimire**[1], **Choden Lama Yonzon**[1], **Yodi Mahendradhata**[4]

1 Faculty of Medicine, Public Health and Nursing, Universitas Gadjah Mada, Yogyakarta, Indonesia, 2 Public Health Youth Society of Nepal, Pokhara, Nepal, 3 Department of Health Behavior, Environment, and Social Medicine, Faculty of Medicine, Public Health and Nursing, Universitas Gadjah Mada, Yogyakarta, Indonesia, 4 Department of Health Policy and Management, Faculty of Medicine, Public Health and Nursing, Universitas Gadjah Mada, Yogyakarta, Indonesia

* mail4sagun@gmail.com

**Data Availability Statement:** All relevant data are available within the paper and its Supporting Information files.

## Abstract

### Introduction

Tuberculosis is one of the leading causes of death worldwide. Diagnosing TB in an early stage and initiating effective treatment is one of the best ways to reduce the burden of tuberculosis. Feasibility of Find cases Actively, Separate safely and Treat effectively (FAST) Strategy helps to improve the early diagnosis of tuberculosis cases among inpatient settings as well as out patient department patients and prevent TB transmission in hospital. This study aimed to assess the feasibility of the FAST strategy, organizational factors, technical factors, barriers and enablers for the proper implementation of the FAST strategy in Nepal.

### Methods

A qualitative study was conducted from April 2019 to August 2019. Data was collected by using focus group discussion, key informant interviews, and client exit interviews. A retrospective research was conducted in different hospitals in Nepal where FAST strategy was implemented. The patients, health care workers, province, district, and National level stakeholders were interviewed. Thematic analysis was used to assess the feasibility as well as barriers and enablers of the FAST strategy.

### Results

Study identified that the 'current setting' of implementation and service delivery arrangement at hospitals were not well arranged as per requirements. The research findings showed hospital ownership is crucial for mobilizing staff and proper space management inside hospitals. Study identified that unavailability of a separate room, limited capacity of GeneXpert machine, irregular supply of GeneXpert cartridge, and insufficient human resources for screening and counseling are the major barriers of FAST implementation in Nepal.

**Funding:** The first author (Sagun Paudel) received funding support as a thesis grant for this study from the World Health Organization Tropical Disease Research (WHO-TDR) Special Postgraduate Programme on Implementation Research at Universitas Gadjah Mada, Indonesia. The funders had no role in study design, data collection and analysis, decision to publish, or preparation of the manuscript.

**Competing interests:** The authors have declared that no competing interests exist.

## Conclusion

FAST strategy is feasible to implement in healthcare settings in Nepal although the technical and organizational factors should be managed to ensure effective function of the strategy as per the approach. Hospital ownership is essential to mobilize health workers, improve client flow system and proper space management for FAST services.

## Introduction

Tuberculosis (TB) is one of the top ten leading causes of death worldwide and the leading cause of death from a single infectious agent [1, 2]. An estimated 10 million people were fell ill and 1.4 million peoples people died of TB in 2019 due to tuberculosis in 2019 [1, 2]. Globally, an estimated 54 million lives were saved through TB diagnosis and treatment services between 2000 and 2017 but there are still large and persistent gaps in the detection and treatment of tuberculosis [1].

Tuberculosis is a major public health problem in Nepal, as it is ranked as the sixth among the leading causes of death in the country [3]. Among the leading causes of death in the country, World Health Organization estimated that 44,000 people develop active TB every year in Nepal. As per the estimation made by WHO, Nepal is still missing around 20–25% i.e.12,000 to 13,000 people who become sick with TB [4, 5]. Early case detection is important to interrupt infection transmission chain within healthcare facilities and community, in addition to reduce suffering time of patient and to reduce the spread of infection from health facilities and untimately at community level.

To identify the missed cases of TB the National Tuberculosis Control Center commenced the FAST (Find cases Actively, Separate safely and Treat effectively) strategy [6]. FAST strategy is an active screening process for the early diagnosis to enable prompt initiation of effective treatment. In FAST approach, the health workers are mobilized to actively search and identify potentially infectious patient in health facilities and the presumptive TB cases are screened, examined and treated. This strategy focused on actively looking for unsuspected TB patients through organized cough surveillance in general medical settings (Fig 1) [7]. The FAST strategy can be used to reduce TB or Drug-resistant tuberculosis (DR-TB) transmission in outpatient and inpatient healthcare settings. A Study conducted in Bangladesh shows that FAST implementation revealed a high frequency of unsuspected TB in hospitalized patients [8]. Similarly, a study conducted in Nigeria shows that the integration of FAST into health care service delivery system improves early detection by reducing the average time of diagnosis [9]. The FAST strategy was initiated in October 2018, as implementation of the this strategy is in the early stage, it is important to understand the feasibility of intervention before scaling up all over the nation. There was no previous study conducted to assess the feasibility of FAST strategy for the diagnosis of TB and infection prevention in Nepal. In this context, this study aimed identify the feasibility of FAST strategy in health facility level and the result of this study aims to suggest National tuberculosis control program for the areas needed to improve. In order to suggest National Tuberculosis control program the areas which are crucial to be improved to strengthen the health facility-based diagnosis and prompt treatment of TB, we aimed to explore organizational, technical and financial factors needed to implement FAST activities in four hospitals.

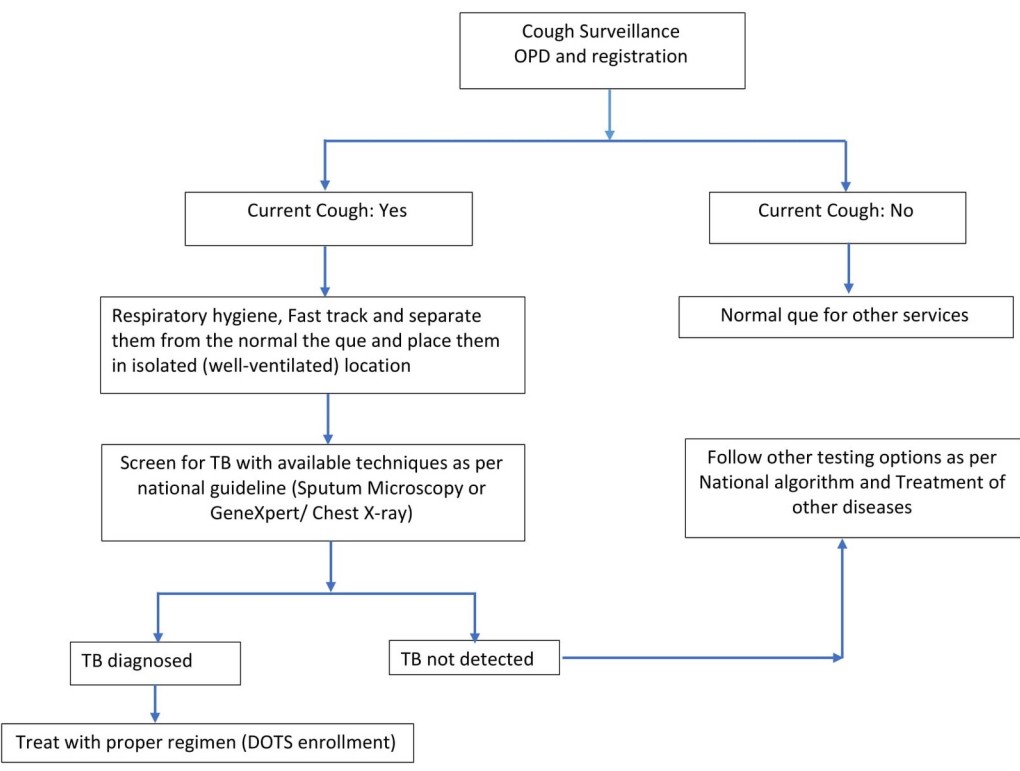

**Fig 1. Find cases Actively, Separate safely and Treat effectively (FAST) approach.**

## Materials and methods

### Study design

This study was an implementation research that used qualitative study design. A total 14 key informant interviews (KII), one focus group discussion and eight clients exit interviews were conducted. A purposive sampling method was utilized to recruit study participants. Respondents were purposively selected from hospitals who were directly involved in the implementation of FAST activities and currently working on FAST activities. The patients who were screened for TB at hospitals were also selected purposively for client exit interviews. Those patients, who attended the out-patient department with chest symptomatic features and screened for TB were selected for the interview purpose. Similarly, district, province & national TB Program focal persons, provincial program manager were selected for KII. Participants for the focus group discussion (FGD) were also selected purposively from National TB Control Centre, implementing partners, & national level stakeholders to achieve the study objective. This study was conducted from April 2019 to August 2019. To ensure the completeness, methodology and reporting the results, the checklist of Standards for Reporting Implementation Studies (StaRI) guidelines was utilized by this research (S1 File) [10].

### Study setting

The FAST is a global fund-supported activity, where various non-governmental organizations are providing technical support to the National TB Control Centre and hospitals to implement this program. The FAST strategy was prepared by the National Control TB Centre in consultation with national-level stakeholders and organizations to initiate this program. Major

seventy-four hospitals were identified as a priority hospitals for FAST implementation however the program was initiated from Dhulikhel hospital and it was further implemented in fifteen different hospitals in Nepal. At the National level, principle recipient of Global Fund, Save the Children was providing technical support to NTC whereas local NGOs were supporting at the hospital level as a partner organization. As FAST approach consists of finding TB patients in outpatient and inpatient setting, health workers actively asked about TB symptoms, separated presumptive TB cases to collect sputum for GeneXpert and diagnosed patients were enrolled for the treatment [7]. The implementation research was conducted in three different settings in Nepal. 1) hospitals in different setting where FAST strategy was implemented, 2) Province and district level health authorities where planning, management, supply related activities are managed and 3) National Tuberculosis Control Centre. To understand the different contexts, data was collected from selected four different hospitals; government hospital, community hospital, teaching hospital and hospital owned by non-governmental organization. The Bir hospital was oldest government hospital in Nepal and the flow of patients is very high as this is major refferal hospital located in Kathmandu. Another study site, Dhulikhel hospital was a community supported hospital in which FAST approach was commenced first time in Nepal. Gandaki Medical College is one of the leading private institution in western region of Nepal and Bayalpata Hospital was located in the rural area of Nepal which was also renowned model hospital operated by NGO in public private partnership model in rural area of Nepal. The purposively selected TB focal person, district and province TB coordinators were interviewed who were primarily responsible to monitor, support and supervise the FAST activities at district and province level. FGD was conducted at the National TB center with National TB program managers and stakeholders who were working to plan, execute, manage, supervise and lead the overall FAST activities at national level.

## Data collection and analysis

The data was collected by using qualitative research instruments. The face-to-face interview with key informants was conducted. The instruments consist of a semi-structured questionnaire and FGD guideline. The Consolidated Framework for Implementation Research (CFIR) domains; inner setting, individual involved, and implementation process were used to prepare the interview tools [11]. The interview guide was developed to determine organizational factors, human resources, availability of services and logistics management, capacity building and knowledge management, information, communication, policy, advocacy related factors for the feasibility of FAST strategy. PI collected data using FGD guideline and Key informant interviews (KII) tool. The KII was used to collect information from service providers, local managers, district and province level program managers and focal persons whereas FGD was conducted at National Tuberculosis Centre with central level program managers and stakeholders who were directly involved in planning, supervision and monitoring of the FASTactivities. The interview was taken with healthcare workers after completion of their regular screening process of patients at hospital. Similarly, client exit interview was taken among patients to explore the client's perspective regarding the FAST approach. Those patients who were screened for TB at hospital by using FAST approach were interviewed after completion of their screening process at hospital. The information was audiotaped using a voice recorder. The recorded information obtained from FGD, KII, and client exit interviews were transcribed into the Nepali language and translated into English. Two research assistants were recruited to assist the principal investigator during data transcription and translation. All information obtained from transcripts, translated documents were re-read, and examined carefully for initial coding. The transcripts were coded, and codes were grouped into manageable categories

and the categories were refined to identify the key theme. Inductive codes were used for writing narratives. To ensure the credibility of the data, the triangulation method was employed.

### Ethics approval

The ethical approval for conducting this research study was obtained from Ethics Committee, Universitas Gadjah Mada, Yogyakarta, Indonesia (Ref. No.: KE/FK/0532/EC/2019) and Nepal Health Research Council (Ref. No. 3038) before starting this study. Approval from National Tuberculosis Centre and permission from hospital administration was taken before conducting study. Written informed consent was obtained prior to the interview by using informed consent form. Privacy and confidentiality of the respondents was well maintained. The participants were ensured that there were no risks of physical, financial and psychological burden. Research instruments and informed consent form was translated into Nepali language.

## Results

### Characteristics of respondents

There were total 27 participants, which included health care providers, hospital managers, district focal persons, provincial team leader, program coordinators and patients. We conducted one mini focused group discussion with national level stakeholders at the National Tuberculosis Centre. Patients were interviewed from four different hospitals (Bayalpata, Bir, Dhulikhel and Gandaki Medical College) to capture the client's perspective regarding the FAST strategy. Key informants, FGD participants, and patients were selected. The characteristics of respondents can be found in Table 1. The responses from the KII and FGDs were grouped into the four major themes as i) implementation context, ii) identified factors for feasibility iii) barriers and enablers iv) way forward for the program implementation.

### Implementation context

The FAST strategy in Nepal was introduced as an infection control measure in selected hospitals where the flow of patients is comparatively high. Respondents added that nowadays FAST also helps to increase case notification of TB at hospitals. National stakeholders described that FAST is implemented as a non-costed intervention as there is no financial implication for program implementation, where TB screening and treatment services are offered at different hospitals.

> "We take FAST as an infection control measure. When we practiced, it also did well with the case notification. [sic]" (National stakeholder, INGO, FGD)

National level program manager indicated that the approach of FAST was easily adopted and implemented at medical college, community hospitals and NGO operated hospitals whereas it was really difficult to initiate at government hospitals. They added that the building structure, service delivery room and waiting room of government hospital is congested and infection prevention measures are not implemented properly as in private hospitals.

> "The OPD has excessively crowded. In the case of Western Regional Hospital, when people come for treatment, it is like competing in a battlefield. [sic]" (TB leprosy officer, government)

It was also observed that the waiting room and flow of patients in government hospital was crowded whereas the flow of patient, client flow system and waiting room arrangement was properly managed in medical college, private hospital and community hospital.

**Table 1. Characteristics of respondents.**

| Character | Categories | KII (n = 14) | Mini FGD (n = 5) | Client exit interview (n = 8) |
|---|---|---|---|---|
| Gender | Male | 6 | 5 | 4 |
| | Female | 8 | 0 | 4 |
| Age | ≥ 40 | 12 | 2 | 5 |
| | < 40 | 2 | 3 | 3 |
| Work experience (in year) | ≥ 17 | 11 | 3 | NA |
| | < 17 | 3 | 2 | NA |
| Education | No Formal Education | 0 | 0 | 1 |
| | Secondary level | 3 | 0 | 3 |
| | Intermediate level | 2 | 0 | 1 |
| | Bachelor | 2 | 5 | 1 |
| | Master | 6 | 0 | 2 |
| | PhD | 1 | 0 | 0 |
| Ethnicity | Dalit | 0 | 0 | 2 |
| | Janajati | 4 | 3 | 2 |
| | Madeshi | 2 | 0 | 0 |
| | Brahmin/Chhetri | 8 | 2 | 4 |
| Respondent type | Health care provider | 5 | 0 | 0 |
| | Provincial and District level officer | 5 | 0 | 0 |
| | National TB Program managers | 0 | 5 | 0 |
| | Hospital manager and administrator | 4 | 0 | 0 |
| | Patients | 0 | 0 | 8 |
| Organization | National TB Centre | 0 | 2 | NA |
| | INGO/NGO | 4 | 2 | NA |
| | WHO | 0 | 1 | NA |
| | Hospital | 9 | 0 | NA |
| | District Health Office | 1 | 0 | NA |
| | | 14 | 5 | 8 |
| Total | | 27 | | |

"It is difficult to work at the government setup. For example, Bharatpur hospital has a very congested and limited area to work. "Therefore, it is difficult to pick the patient, compared to the Teaching hospital and Achham hospital. [sic]" (National stakeholder, FGD)

Participants explained that the implementation setting was not readily structured as per the requirements, however they strongly argued that the hospital setting. Service operation process can be easily modified as per FAST guideline to implement the FAST approach and physical space can be managed inside the hospitals.

"It's all about the administrative arrangement, there was no such a special technical problem in implementing sites so we can easily implement if ownership can be taken by respective hospital. [sic]" (TB leprosy officer, government)

Based on their experiences, participants identified numerous benefits of the FAST strategy. They explained that the number of presumptive cases and case notification was increased in the selected hospitals. FAST increases a sensitization among healthcare providers for self-protection as well as symptomatic patients are prioritized to examine faster.

"It has helped to lessen the transmission cycle and we noticed that case has been increased after the FAST strategy in comparison to before. [sic]" (hospital director, NGO operated hospital)

"Sometimes the doctor could not identify the TB case in their routine screening process and the possibility of losing the case is high but now FAST can detect the case early. [sic]" (Nursing administrator, medical college)

Respondent indicated that the approach of the FAST strategy was not implemented as per the expectation at the initial stage of program implementation. The preliminary results show that the TB case identification was not as expected. Both national level program managers and hospital managers raised a common issue on staff mobilization and space management.

They added that screening of cases at the hospitals was performed per protocol, however the separation of suspected TB patients was not properly done because there was no adequate space to keep patient separately and there was no individual assigned for TB screening.

As the FAST approach consists of series of activities in the hospital setting, our study identified that the actively screening, counseling, and sputum examination was conducted as per the framework of FAST strategy. However separation of the patient from the queue and place them in the isolated or well-ventilated room was not followed as per the requirement of the program. It was found that the diagnosed cases were properly enrolled for DOTS. As per the FAST framework, the main problem was identified on the separation of patients. This study identified that the treatment of diagnosed TB cases was initiated timely and the patients were enrolled at DOTS therapy within hospitals.

## Identified factors for the feasibility of FAST

Various factors are identified as an essential factor for the proper implementation of the FAST approach in hospital settings. Different organizational perspectives, such as ownership, separate room, and interdepartmental coordination were associated with perceived facilitating factors for FAST implementation. This study identified that the FAST strategy can be implemented undoubtedly if all these things are working and managed.

## Hospital ownership

Participants repeatedly stressed that strong hospital ownership is a fundamental factor for staff mobilization, arrangement of the client flow system, registration desk, and waiting room. Similarly, hospital ownership is required for overall program implementation, monitoring, recording, reporting, and evaluation of the program. We identified that the hospital ownership was established in a community hospital and NGO-operated hospital to mobilize local resources for FAST whereas the government hospital didn't realize it.

*"We thought it is the hospital's responsibility. They should take ownership to control the infection and separate patients. [sic]" (National Stakeholder, FGD)*

## Room availability

From KII and Client's exit interview, it was found that the availability of a separate room or confidential space impacted on the effectiveness of FAST services. Many health care providers at implementing hospital defined the importance of separate room for counseling and screening to maintain the privacy and confidentiality of patients. Respondents identified that separate waiting room is essential for patients, and it should be well ventilated to prevent infection

transmission. Arrangement of FAST services in a limited space overcrowded with patients for screening was identified challenges in government hospital whereas it was separately arranged in community hospital and NGO operated hospital.

*"In reality, the density of patients is high in hospitals, and it is difficult to identify the case. The patients are roaming around the hospitals for their turn. Meanwhile, the TB patient is spreading the TB around. So, to prevent infection transmission separate waiting room is essential for suspected patient. [sic]" (TB leprosy officer, government)*

Many participants argued that the arrangement of FAST desk or specific space for TB screening near the registration room at the hospital will facilitate to improve the screening process among all patients. One of the respondents expressed that separate booth for sputum collection is required to prevent the transmission of infection from patient to the environment and people around the hospital.

*"The client goes outside and starts coughing without caring for the people around them. If we have the particular sputum booth, then we will use that booth to produce and collect sputum only. [sic]" (District program coordinator, NGO)*

Health service providers and patients talked about the arrangement of the FAST services; screening, sputum collection, counseling and treatment in the same floor in the hospital will make easier for patients. Our observation also suggests that services for screening, counseling, and sample collection all at one place makes easier for patients to receive a service timely. It was observed that all services for FAST process was arranged in same floor in NGO operated hospital whereas in other hospitals services were given from different floor. Many healthcare providers mentioned that use of FAST stamp helps to prioritize the patients in the hospital to provide quick services to the patients. Respondents argued that when OPD ticket is stamped on the spot for FAST examination, the patient is checked up more seriously and carefully after seeing the stamp. Some participants argued that implementation of FAST activities in a hospital setting is a collaborative task of administrative staff and healthcare providers, specially highlighted by health workers who are working in government hospital. They reported that only a few people and volunteers cannot complete all these activities, for that inter-departmental coordination between laboratory, wards as well as registration department is required. Majority of respondents also suggest that activities of FAST should be integrated to the existing committee on infection prevention to manage it properly as FAST approach is also working to prevent hospital-based infection prevention. They explained that the FAST activities and hospital infection prevention committee are not working jointly until now. Few respondents from NGO hospital highlighted the important role of patient navigator *"birami sahayogi"* to helps patients to move from one department to another department. Birami sahayogi is the administrative staff mobilized to support patients for searching service rooms inside hospitals. It was found that the NGO operate hospital initiated a novel concept of patient navigator whereas there were no such patient support staffs in other hospitals. The patient navigators are the support personnel mobilized at hospital to provide information to patients and guide them to move from one room to another room to get timely services.

*"Our hospital has a unique system of patient navigators, we say "birami sahayogi" which I think is necessary for other hospitals where the patient flow is high and to implement a FAST strategy to manage properly. [sic]" (Hospital director, NGO operated hospital)*

It was found that, the hospital dress like using apron by NGO staff during their duty hour at hospital may facilitate for better communication with patients. Respondents emphasized that dedicated staff for screening and counseling of patients is necessary for proper management of FAST activities in hospitals.

*"We have hired two staffs in each 15 hospitals as a full-time employee. After that, the number of FAST case and presumptive has increased. [sic]" (National stakeholder, INGO, FGD)*

Similarly, few respondents argued that the existing staff of the hospital should be mobilized for the sustainability of the program. Some respondents reported that there is no workload for healthcare providers to implement FAST activities in the hospital. However, some described that there is a workload for laboratory staff and patient's navigator because the patient had to visit different rooms. Majority of respondents explained that there is a positive role of incentives to motivate healthcare providers. One responded added that the hazard allowance provided by the government for laboratory staff should be distributed on time to motivate them.

*"NRS. 50 per case was provided by the government as a hazard allowance for the laboratory staffs it motivates them. [sic]" (District program coordinator, NGO)*

## GeneXpert and DOTS service

GeneXpert service was identified as a key factor to implement the FAST approach at hospitals. Respondent highlighted that the GeneXpert should be prioritized to reduce the duration of diagnosis time. Respondents explained that the limited capacity of the GeneXpert machine is one of the challenges to test all samples and provide results in the same day. Respondents stated that they can only test maximum twelve samples in an office hour. So, if the number of TB suspected patients increased, the reporting time will be prolonged until the next day. The basic understanding of FAST is to provide a report and treatment to the patients within a day, however due to prolonged time on laboratory report, the service cannot provide in a same day when the patient's flow is high. Respondents highlighted only two technicians were available at National TB Centre to machine maintenance and repairment all around the country, so it takes long time to maintenance, calibrate and repair machine. This study reveals that there should be well managed supply and storage of GeneXpert cartridge. Similarly, it was found that the cartridge is temperature sensitive and it should maintain 2–28 degrees Celsius for its proper storage.

Another identified issue was the availability of DOTS treatment center in the same health facility will be more beneficial to the FAST approach. Respondent argued that arrangement of all services; screening, diagnosis and treatment in the same hospital makes it easy for patients and it will help to prevent the chances of missing patients when the patient is referring from one hospital to the other institutions for DOTS treatment.

## Capacity building

This study reveals that training and orientation were required for service providers and hospital administrative staffs. Similarly, National stakeholders also described that training and orientation is the key factor to FAST implementation and motivate hospital administrative staff to initiate FAST in the hospital. It was reported that the existing modular training on TB and infection prevention is sufficient to build the capacity of local health workers. It was identified that onsite coaching, supervision, monitoring and monthly review meeting plays a positive role in the capacity enhancement of healthcare providers.

## Counseling tool

The results showed absence of uniformity on counseling counseling among chest symptomatic patients. This study identified that group counseling for patients will be beneficial during the waiting time of patients.

> *They [Health worker] told FAST. FAST but I don't know what it is. . . ha ha ha.. [sic]" (Client exit interview, male, 32-year, bachelor level education)*

Another identified issue was the counseling tool is important to provide complete and concise health information. Participants added that there is a no flex board, information booklet, wall chart, pamphlets and poster to convey adequate information to patients.

## Advocacy

This study found that the advertisement of FAST activities plays an important role in the proper implementation of the program as well as the linkage of the FAST activities with community TB awareness program will be beneficial to increase the utilization of services.

> *"This type of FAST services should be advertised, and every chest symptomatic patient should be screened. [sic]" (Client exit interview, female, 28-year, master level education)*

## Policy formulation

National level stakeholders stressed the importance of policy formulation. A lack of clear policy is the main challenge to convince hospital administration to implement activities inside the hospital. FGD Participants also pointed out to conduction of advocacy activities among hospital administrative to make a clear understanding of FAST and its importance.

> *"It needs to be discussed at the policy level. If policy directs hospitals to segregate the coughing patients, it won't bring a big challenge for proper implementation. [sic]" (National stakeholder, INGO, FGD)*

Many respondents said that FAST is an administrative intervention that can easily adopted within the normal service delivery approach of hospitals and routine TB screening and treatment protocol. National TB Centre provides services free of cost to all peoples for Sputum Microscopic examination, GeneXpert and TB treatment services, so there is no financial burden to hospitals. So, this study found out that a separate budget for FAST activities is not required for diagnostic procedures and TB treatment. However, the logistics and equipment should be regularly supplied as well as proper incentives to the healthcare providers is required to motivate them.

## Barriers and enablers

Unavailability of a separate room, limited capacity of GeneXpert machine were the commonly reported barriers. Similarly, lack of clear policy, irregular supply of cartridge for GeneXpert, insufficient human resources for screening and counseling in high OPD flow hospital, delay in laboratory result due to a high volume of samples are also identified barriers from service delivery side. Inadequate counseling, no distribution of Information, education and communication (IEC) materials such as booklets, wall charts, pamphlets and information sheet as well as no advertisement of the FAST approach were reported barriers from patient side. Similarly, few enabling factors were highlighted by participants. Free TB service and equipment supplies

from government is one of the major enabling factors. Multi-stakeholder engagement and partnership between various organizations, regular supervision, monitoring, and onsite coaching help to create an enabling environment to implement FAST activities locally. Some hospital administrators and healthcare providers reported that dedicated staff for screening and counseling is also an enabler for better service. Provision of hazard allowance for laboratory staff, availability of DOTS service in the same hospital was also noticed as an enabling factor.

## Way forward for program implementation

Participants specifically mentioned that the FAST strategy is feasible to implement in the context of Nepal however, they highlighted that the availability of laboratory facility, space management, hospital ownership and dedicated staff to screen and counsel the suspected patients should be ensured. Several participants suggested that the preliminary result of the FAST implementation is very good for infection prevention and to increase case notification.

> *""It is a very basic matter,. . .. Space management, a supportive laboratory staff, GeneXpert and one dedicated staff is enough to implement FAST. [sic]" (Hospital focal person, community hospital, KII)*

They clearly emphasized that FAST should be implemented in every hospital. Healthcare providers from different hospitals explained that there is no complex procedure to implement FAST in the hospital setting. They added that FAST is easy adoptable however the enabling environment at implementing site is required. This research identified that administrative management is an important thing to make FAST feasible in hospitals.

## Discussion

This study is the first of its kind in Nepal to explore the current context of FAST implementation in Nepal. Our study explored the view of frontline health care providers and managers on the current implementation status, barriers and enablers in the FAST approach to early diagnosis of TB and infection prevention. As a feasibility study, our study analyzed the different aspects of organizational, technical, operational and financial aspects of FAST [12]. Our study revealed that hospital ownership and administrative support are fundamental factors in the implementation FAST approach effectively. The structural arrangement is a key factor to prevent infection transmission in a hospital setting, so to make FAST as a sustainable approach for infection prevention, infrastructure, and patient queue management is required [13]. Our research reveals that a separate well-ventilated room should be managed to prevent the infection transmission to other patients from suspected patients, this result is similar with the study conducted in Nigeria and recommended by the WHO policy on TB infection control [14, 15]. This study highlighted that policy formulation is necessary for the FAST strategy functioning, so the hospitals will be obliged to implement the process in healthcare facilities [15]. Effective coordination between health care providers, administrative staff, other hospital staff, and inter-departmental coordination is necessary to optimize the benefits of the program. Concurring with the findings of our study, findings from Bangladesh show that dedicated implementation team efforts are needed for better performance at the hospital level [8]. Our study explored that the screening approach increases the workload at laboratory to test all the samples which were also similar to the study conducted in Tanzania [16]. The use of the GeneXpert machine for sputum examination will reduce the turn-around time, for this, a continuous uninterrupted supply of cartridge is required, and the capacity of the laboratory should be increased. Our study identified that the GeneXpert module failure and delay in maintenance, which is

also shown in Bangladesh [8]. The result is similar to the study conducted in Bangladesh which identified that the laboratory capacity is important for the sustainability and scalability of the program [8]. We also identified few issues in information and health education materials as a gap in hospital settings such as display of flex board, information booklet, wall chart, pamphlets and poster inside the hospital to convey adequate information to patients will increase the understanding of FAST approach and utilization of services among patients. This clearly calls program managers and hospital administrators to design information education and communication (IEC) materials and advertisement of FAST process and TB screening.

This study reveals that FAST is a non-costed intervention as there is no additional financial burden for hospitals. So, this study found out that FAST can be implemented from the available resources and routine supplies from National Tuberculosis Control Centre. As similar to the finding of our result this is particularly relevant in Nigeria where research identified that FAST can be easily implemented with minimal resources [9]. This study identifies that the proper space or separate room for segregate the suspected patients inside the hospital is a major barrier to the proper implementation, this is consistent with findings in Nigeria [9]. Research paper described that patient flow, separation, airborne isolation rooms, air disinfection, and respiratory protection always plays a vital role in TB patient management [8]. As found in this study, limited capacity of GeneXpert machine, irregular supply of cartridge for GeneXpert, and timely maintenance of the machine were considered as barriers for uninterrupted delivery of laboratory services, a similar finding was reported from the study conducted in Nepal [17]. The results of this study indicate that the FAST approach supports to evaluate presumptive TB patients actively, separated and enrolled in the DOTS center when required [18]. As same as the result of this study, a large number of undiagnosed cases of TB can be identified by systematic screening of the OPD patients in Pakistan [19]. The major benefit of the FAST strategy is early diagnosis and prevent transmission at the hospital which is reported by a similar study in Russia that FAST plays an important role to limit the possible consequences of MDR tuberculosis [20].

We could not access all the hospitals and all healthcare providers, managers and focal persons because of limited time period of study however the information and result obtained from this result may be useful for planners and stakeholders to further improvement in similar context and settings. Few patients who performed the sputum examination in the hospital were not returned to receive laboratory results, so it was challenging to take a client exit interview in government hospital however interview was taken with other patients who perform sputum examination after the FAST approach.

Our study showed that consolidated framework for implementation research (CFIR) is very useful for collecting qualitative data in low resource because it helps to cover the various aspects of program implementation. We were able to capture comprehensive and organized information by using CFIR. As CFIR provided different constructs of questionnaires, the complete information during FGD and KII was collected easily. Sometimes, few constructs such as the inner setting and client's perspective were scattered, and it makes it difficult to analyze the constructs separately. However, our study and experience suggest that using CFIR improves our research quality, especially on tools designing and capturing complete information including barriers and enablers of the program implementation.

## Conclusions

This study provides valuable information for implementation and feasibility of FAST strategy in Nepal. The findings of this research conclude that the implementation set-up and service delivery mechanism at hospitals is not yet well practiced as a routine process for the FAST

approach however it can be managed if hospital administration takes responsibility to institutionalization this approach. Various factors were identified such as hospital ownership, separate room, interdepartmental coordination, the establishment of FAST desk, sputum collection booth is identified as an essential factor for the proper implementation. Similarly, the Availability of GeneXpert services, regular supply of cartridge, maintenance of GeneXpert machine and availability of DOTS treatment center in the same hospital are identified technical factors. From the results obtained, it is suggestive that FAST strategy is feasible to implement for early diagnosis of TB and infection prevention in healthcare settings however the identified factors should be managed to ensure effective program implementation.

## Supporting information

**S1 File.**
(DOCX)

**S2 File.**
(XLSX)

## Acknowledgments

We would like to express our gratitude to WHO/TDR postgraduate scholarship program/ Universitas Gadjah Mada, Indonesia for providing scholarships. We would like to thanks to National TB Center, Mr. Sagar Prasad Ghimire, Dr. Ashish Shrestha, Dr. Suvesh Kumar Shrestha for their valuable support and advice during research. We also appreciate to Dr. Bhim Singh Tinkari, Dr. Sagar Kumar Rajbhandari, Dr. Bikash Gauchan and all the respondents who participated in this research study.

## Author Contributions

**Conceptualization:** Sagun Paudel, Retna Siwi Padmawati, Yodi Mahendradhata.

**Data curation:** Sagun Paudel.

**Formal analysis:** Sagun Paudel.

**Investigation:** Sagun Paudel.

**Methodology:** Sagun Paudel.

**Project administration:** Sagun Paudel, Ashmita Ghimire, Choden Lama Yonzon.

**Resources:** Sagun Paudel.

**Software:** Sagun Paudel.

**Supervision:** Retna Siwi Padmawati, Yodi Mahendradhata.

**Validation:** Sagun Paudel, Ashmita Ghimire, Choden Lama Yonzon.

**Writing – original draft:** Sagun Paudel.

**Writing – review & editing:** Sagun Paudel, Retna Siwi Padmawati, Ashmita Ghimire, Choden Lama Yonzon, Yodi Mahendradhata.

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
