## [Decision Letter · Decision Letter 0]

3 Dec 2020

PONE-D-20-20684

Feasibility of Finding Actively Separating and Treatment (FAST) Strategy for Early Diagnosis of TB in Nepal: An Implementation Research

PLOS ONE

Dear Dr. Paudel,

Thank you for submitting your manuscript to PLOS ONE. After careful consideration, we feel that it has merit but does not fully meet PLOS ONE’s publication criteria as it currently stands. Therefore, we invite you to submit a revised version of the manuscript that addresses the points raised during the review process.

We look forward to receiving your revised manuscript.

Kind regards,

Shyam Sundar Budhathoki, MBBS, MD, MPH

Academic Editor

PLOS ONE

Journal Requirements:

Reviewers' comments:

Reviewer's Responses to Questions

**Comments to the Author**

1. Is the manuscript technically sound, and do the data support the conclusions?

Reviewer #1: No

Reviewer #2: Yes

2. Has the statistical analysis been performed appropriately and rigorously? 

Reviewer #1: N/A

Reviewer #2: N/A

3. Have the authors made all data underlying the findings in their manuscript fully available?

Reviewer #1: Yes

Reviewer #2: No

4. Is the manuscript presented in an intelligible fashion and written in standard English?

Reviewer #1: Yes

Reviewer #2: Yes

5. Review Comments to the Author

Reviewer #1: Feasibility of Finding Actively Separating and Treatment (FAST) Strategy for Early Diagnosis of TB in Nepal: An Implementation Research

PONE-D-20-20684 Reviewer comments

Overall critique

FAST (Find cases Actively, Separate safely and Treat effectively) is a refocused TB infection control strategy, originally designed to reduce TB transmission in inpatient healthcare facilities (where TB treatment could be initiated based on rapid molecular diagnostic testing). Results from a large implementation trial of FAST in Peru are pending but FAST is being scaled up in various settings. This article presents an analysis of FAST implementation in different healthcare contexts in Nepal using qualitative research methods. While data analyzing implementation of an administrative TB transmission prevention strategy such as FAST hold great value, in its current form the manuscript does not adequately discuss the critical component of early initiation of effective treatment (central to FAST), lacks adequate detail regarding the qualitative methodology used (for example, CFIR is mentioned in the discussion as a framework used to guide data collection but not mentioned earlier) and does not present a sufficiently nuanced discussion of the themes identified (nor does it even specify whether these themes arose from inductive vs deductive analysis). Although one of the major strengths of the study is the wide range of stakeholders that were interviewed, the analysis does not convey this as it is often unclear which stakeholders and settings each descriptive theme or sub-theme refer to or discuss differences between these contexts and perspectives which is a missed opportunity. Finally the conclusions drawn do not represent assertions that can be made based on these data and should be revised.

Specific comments by section

Title – this needs revision as it does not encompass the full definition of FAST (see above first line of comments) as FAST is not merely about early diagnosis but early diagnosis coupled with effective treatment

Abstract

Needs full definition of FAST as above

Sentence beginning line 23 needs to incorporate importance of effective treatment and would not focus on OPD here as FAST was initially designed for inpatient settings

Introduction last three sentences read in a disjointed fashion – suggest revising

Methods (would remove materials as not relevant to this study type) – state prospective vs retrospective study

Focus rather than focused group discussion

Methods section is too brief – suggest shortening introduction to enable addition of text re: healthcare setting/country context and analytic methodology

Results are not well organized and primarily consist of lists, suggest revising

Conclusion – sentence about hospital ownership is important but requires a bit more explanation e.g. hospital ownership (as opposed to?)

Introduction

Update data in first sentences based on WHO 2020 report

Line 56- unclear relevance of latent TB for this study- suggest removing

Line 57 – 44% of cases – I think this refers to active rather than latent TB? Would move or remove since you later discuss Nepal specific data.

Line 65 – would rephrase incidences as people who become sick with TB

Lines 66-68 – unclear why you are discussing the 2013-2015 data since this study was done subsequently

Line 68-69 – this statistic repeats the earlier statistic re: 12-13 000 cases missed/year

Lines 70-71 – suggest rephrasing

Line 72 – as mentioned above please define FAST as per Barrera, Nardell IJTLD reference and cite this

Line 74 – early diagnosis to enable prompt initiation of effective treatment

Line 75-76 - would replace chest symptomatic with potentially infectious patient

Line 77-79 – would remove as FAST has not been proven to be the best strategy as stated

Line 79-81 - would state that FAST was initially designed for implementation in inpatient settings because of the central importance of early effective treatment playing a major role in decreasing transmission

Line 84-85 – would remove by healthcare workers and cough officers as optimal approach for implementing FAST has not yet been established

Lines 92-94 – this describes the study results and so should not be in the introduction

Methods

Line 98 – revise sentence re: study design

Line 99 – focus rather than focused group

Lines 100-101 – please briefly describe purposive sampling method/describe how participants were recruited

Lines 106-107 – difference between 1) and 2) is not entirely clear- please clarify.

Lines 108-110 – this answers some of the prior question re: purposive sampling, would move

Line 114 - you say similarly but test appears to be more in keeping with ‘In contrast..’ and you mention this is a community rather than referral hospital so unclear what is similar?

Line 119-120 – describe how these stakeholders were selected/sampled and from which healthcare contexts?

Line 124 – what qualitative research instruments? Explain this? Also you have not defined KII earlier.

Line 125 – KII not defined before

Line 128 – clear that patient interviews were exit interviews- how and when were interviews with health workers performed?

Line 129 – how were these instruments developed?

Lines 133- 137 – requires more detail e.g. inductive versus deductive analysis and reference for thematic analysis e.g. Braun and Clarke

Context of how FAST implementation occurred in the country and at your sites should be described in the methods, including how was responsible for implementing FAST

Also need description of what FAST implementation means e.g. screening all patients with cough vs other respiratory symptoms? Did all get Xpert? Did some get chest x-ray?

Results

Characteristics – should report numbers for all types of data including focus groups. For KIIs, would not list districts but FAST implementing sites as you do for the patients

I recommend a paragraph describing how and which themes were derived before describing findings, currently unclear if these subheadings are the themes based on inductive versus deductive analysis

Line 164 – what does good result re: case notification mean?

Line 165 – what does implemented as a non-costed intervention mean?

Lines 168-179 – see above comment re: context of how FAST implementation occurred in the country and at your sites should be described in the methods

Line 180 – when you say respondent, should specify from which participant group

Line 186 – crowded not crowed

Line 197-199 – clarify you mean this could be changed with administrative buy-in/support

Lines 217-218 – please clarify what this means: case identification was not as expected and low number of patients were screened for TB

Line 223 – remove stigmatizing term suspected, also in line 272, 284

Line 232 – what about initiation of treatment?

Line 249 – which hospital does this quote refer to?

Lines 289-300 – is this all hospitals/some hospitals? Which respondents?

Line 299 – who were these patient navigators?

Line 385 – again explain non-costed intervention? Integrated into routine care?

Line 400 – what are IEC materials?

Discussion

Line 437-439 – unclear relevance of this citation?

Line 461-463 – similar to earlier, unclear relevance of this citation?

Line 471-473 – this contradicts what you say re: need for space and dedicated staff?

Line 485 – repetition with earlier section at line 455 re: Xpert

Lines 489-490 – this citation does not refer to FAST so clarification needs to be made

Line 493 – rephrase, I think you mean prevent transmission?

Lines 506-517 – this is the first mention of CFIR (should cite Damschroeder 2009) and this discussion is inadequate. If CFIR was used to guide data collection this should be in the methods and the themes described should discuss CFIR domains e.g. inner/outer setting more clearly

Conclusion

Lines 522-523 - need to rephrase

Lines 533-534 – you do not provide information that FAST reduces transmission or even provide data about number of cases diagnosed so cannot conclude this

Supplemental data link does not work

Reviewer #2: Introduction

Line no 25: Please remove visiting

Line no 25: It should be Tb infections..

Line no 28: “This study also explored barriers and enablers that

influence the proper implementation of the FAST strategy in Nepal”.. it sounds much better if we rephrase it as “In addition, this study also…”

line no 36: change was to were.. change to as per the requirement

line 44: please rephrase the sentences “freely available of services”

line no 55: does “ill” mean people were sick from TB or were they just infected with TB bacilli? Not all TB infected people become sick from TB.

line no 56: “In 2017, 1.6 million people were died” should be rewritten was 1.6 million people died from TB.

line no 58-60: please rephrase the sentences.

Line no 64-65: “As per the estimation made by WHO, Nepal is still missing around 12,000 to 13,000 incidences”—is this the detection gap? Incidences or cases?

Line 66-69: please rephrase the sentences as its confusing.

Line no 71: spread in health facility or community?

Line no 106: change to different settings

Line no 130-131: PI collected data using FGD guideline and KII tool.

Line no 139: delete be.

Line no 151: A total of fourteen health care providers,: was it 14 health care providers or total participants were 14?.. rephrase as there were total 14 participants, which included..

Line 163: “high OPD patient flow hospitals.” Please rephrase this sentences

Result section

The result section needs to be rewritten. Its quite difficult to comprehend some sentences.

General comments

Proof reading has to be done to check for grammar and sentencing.

6. PLOS authors have the option to publish the peer review history of their article (what does this mean?). If published, this will include your full peer review and any attached files.

Reviewer #1: No

Reviewer #2: No

---

## [Author Response · Author response to Decision Letter 0]

9 Feb 2021

Respected Sir,

Thank you for the opportunity to revise our manuscript entitled “Feasibility of Find cases Actively, Separate safely and Treat effectively (FAST) strategy for Early Diagnosis of TB in Nepal: An Implementation Research” PONE-D-20-20684. We are thankful to both reviewer 1 and reviewer 2 for the constructive comments and feedback on our manuscript. 

We feel that the suggestions received from reviewers are insightful and helped us in improving the manuscript. We have made attempt to fully address these comments and incorporate the feedback in the revised manuscript and believe our revised manuscript represents a significant improvement over our initial submission. Please find attached a response to reviewers for your kind consideration. We hope that you find our responses satisfactory.

In addition, I would like to kindly inform you that the available data sets for this manuscript are also attached in supporting files.

Many thanks again. 

Sincerely,

Sagun Paudel, MPH

---

## [Decision Letter · Decision Letter 1]

18 Jun 2021

PONE-D-20-20684R1

Feasibility of Find cases Actively, Separate safely and Treat effectively (FAST) strategy for early diagnosis of TB in Nepal: An implementation research

PLOS ONE

Dear Dr. Paudel,

Thank you for submitting your manuscript to PLOS ONE. After careful consideration, we feel that it has merit but does not fully meet PLOS ONE’s publication criteria as it currently stands. Therefore, we invite you to submit a revised version of the manuscript that addresses the points raised during the review process.

The reviewers felt that although the previous points were largely addressed, some concerns remained. They felt that the methodology and study design should be better justified, and some of the variable better defined. They also felt that some of the conclusions were not directly supported by the results of the study and should be revised. Their comments can be viewed in full, below.

We look forward to receiving your revised manuscript.

Kind regards,

Natasha McDonald, PhD

Associate Editor

PLOS ONE

Journal Requirements:

Reviewers' comments:

Reviewer's Responses to Questions

**Comments to the Author**

1. If the authors have adequately addressed your comments raised in a previous round of review and you feel that this manuscript is now acceptable for publication, you may indicate that here to bypass the “Comments to the Author” section, enter your conflict of interest statement in the “Confidential to Editor” section, and submit your "Accept" recommendation.

Reviewer #3: (No Response)

Reviewer #4: (No Response)

2. Is the manuscript technically sound, and do the data support the conclusions?

Reviewer #3: Partly

Reviewer #4: Partly

3. Has the statistical analysis been performed appropriately and rigorously? 

Reviewer #3: I Don't Know

Reviewer #4: No

4. Have the authors made all data underlying the findings in their manuscript fully available?

Reviewer #3: Yes

Reviewer #4: Yes

5. Is the manuscript presented in an intelligible fashion and written in standard English?

Reviewer #3: No

Reviewer #4: Yes

6. Review Comments to the Author

Reviewer #3: Overall, it is an important and needed topic for the health system especially on the 'How'

I have minimal competency in qualitative design to do justice to the paper-especially on the methodology and analysis

There is a need for better justification, gap analysis in existing knowledge and practice and alignment with a clear study objectives.

It will be helpful to describe better the variables considered under organizational, technical & financial factors in the data collection and analysis section-line 123

Line 151-157 should all be part of the methodology

The conclusion: line 529-532 especially linking the conclusion with early diagnosis is not supported by the study.

There is a need for better use of scholarly tone and editing

Reviewer #4: Comments to the author:

The authors address an important subject in TB transmission prevention and control, such as implementing FAST strategy. As long as the process is complex and requires input of different governmental and non-governmental associations in addition to healthcare facility members, the study covers most of the aspects that might be associated with difficulties of proper functioning of the strategy. Even though the sample size is small it seems to be enough to notch up gaps which should be considered and covered by National TB control program. It would also be interesting to follow-up the process over the time.

Please find the recommended changes in the manuscript below.

Abstract

Point 1: line 24 OPD – change to full wording

Point 2: line 35 – rephrase “study identified that the current setting of implementation…”

Point 3: line 37-38 – rephrase – “…showed that hospital ownership is crucial/is important for mobilizing staff (not staffs)….”

Point 4: line 38 – remove “that”

Point 5: line 40 - cartridge not cartilage

Point 6: line 40 – manpower (not fair with women) = human resources

Point 7: line 44 – rephrase “… effective function of the strategy”

Introduction

Point 8: line 50 – rephrase, it is not top ten it is among top ten, caused by single infectious agent, update WHO report date – 2017 to the newer version.

Point 9: line 52 - remove “which is steady in 2018”

Point 10: line 57 – add “… among the leading causes of death in the country”

Point 11: line 64 - rephrase – “Early case detection is important to interrupt infection transmission chain within healthcare facilities and community, in addition to reduce time of patients` suffering.”

Point 12: line 63 – NTC needs full wording

Point 13: line 65 – change activity to process.

Point 14: line 68-70 - remove sentence starting with “Fast was initially…”

Point 15: line 71 – figure 1 – figure misses “No” in the upper right option of current cough and the options of two conditions such as: 1. Signs and symptoms – yes / chest xray –normal; 2. Signs and symptoms –no / chest xray abnormal; - what happens in such situations?

Point 16: line 73 – define DR- TB

Point 17: line 77 – change early diagnosis to early detection

Point 18: line 82-88– rephrase to – “In order to suggest National Tuberculosis control program the areas which are crucial to be improved to strengthen the health facility based diagnosis and prompt treatment of TB, we aimed to explore organizational, technical and financial factors needed to implement FAST activities in four hospitals.”

Materials and methods

Point 19: line 97 – define criteria for “purposively selected patients” – meaning they were selected based on…?

Point 20: line 92 – include FGD abbreviation with focus group discussion

Point 21: Study setting – start paragraph with line 110 – “The FAST is a global fund-supported activity, where various non-governmental organizations are providing technical support to the …..” continue with line 106, “The FAST strategy was prepared by ….”

Point 22: line 115 change “The FAST approach” to “As FAST approach”

Point 23: line 117 change “separated presumptive TB cases to collect sputum for GeneXpert and diagnosed patients were enrolled for the treatment”

Point 24: line 120 – difference or different?

Point 25: line 125 – remove part of the sentence starting from “because these hospitals…”

Results

Point 26: line 188 remove “purposively”

Point 27: line 190 – define groups clearly, eg. i) implementation context, ii) identified factors for feasibility iii) barriers and enablers iv) way forward for the program implementation

Point 28: Table 1 – Patient who is 40 years old, is over 40 or below 40? Define age categories like ≥40 and <40 at least. Or 18-39 / 40 and up… Same goes to work experience length.

Point 29: line 197 – respondent or respondents?

Point 30: line 199-200 rephrase

Point 31: line 219 – change “So, it is difficult” to “Therefore it is difficult to pick the patient, compared to the Teaching hospital and Achham hospital.”

Point 32: line 222-225 rephrase whole paragraph “…and physical space can be managed inside the hospitals.”

Point 33: line 228 –include “[sic]” at the end of every quote, indicating there might be grammatical problems and you are not interfering with corrections.

Point 34: line 231 – change “… increased in implementing” to “… increased in the selected hospitals.”

Point 35: line 232 remove chest and change “… are prioritized to be examined faster.”

Point 36: line 242 – “as per the expectation at the initial stage…”

Point 37: line 243 – “…results show that the TB…”

Point 38: line 246 – rephrase – “…at the hospitals was performed per protocol, however…”

Point 39: paragraph lines 250-259 – shorten too many repetitions. Needs just a few sentences.

Point 40: lines 272- 275 – unclear, please rephrase, avoid repetition of the specific words.

Point 41: line 282 – change stressed to defined

Point 42: line 286 – rephrase “… limited space overcrowded with patients…”

Point 43: line 311 – Respondent or respondents?

Point 44: line 324 – explain the meaning of “birami sahayogi”.

Point 45: line 327 – rephrase “… support staffs mobilized..” to “… support personnel mobilized…”

Point 46: line 348 – remove “... so it was not necessary for FAST”, but would be better to rephrase the whole sentence.

Point 47: line 361- result waiting time is turn-around time or reporting time. Change “unitl the next day.”

Point 48: lines 362-363 – rephrase to “… the patients within a day, however…”

Point 49: line 375 – add “the” – “… to the other institutions...”

Point 50: line 386 – rephrase – “The results showed absence of uniformity on counseling….”

Point 51: line 408 – rephrase voiced to pointed out. Remove “is required” at the end of the sentence.

Point 52: line 425 – manpower = human resources / work force

Point 53: line 428 – full wording for IEC.

Point 54: line 438 - “… DOTS service …. was also noticed as the enabling factor”

Point 55: line 441 - change stressed to highlighted

Point 56: line 453 – maybe easy adoptable instead of “doable and possible”

Discussion – Overall, this section is too big. There are many unnecessary parts, please shorten considerably. Do not repeat results as they already are in the section above.

Point 57: line 464 “… support are fundamental factors…”

Point 58: line 466 – “… prevent infection transmission in a hospital setting…”

Point 59: line 472 – “… is necessary for the FAST strategy functioning, so the hospitals will be obliged to implement the process in healthcare facilities.”

Point 60: line 485 – “… delay in the maintenance..,”

Point 61: line 502 – “…barrier to the proper…”

Point 62: line 507 – “… and timely maintenance…”

Point 63: line 508 – “… were considered as barriers for…”

Point 63: lines 517-520 – remove and start “We could not access…”

Point 64: lines 529 -533 – remove and start “Our study showed that CFIR….”

Conclusion

Point 65: line 542 – rephrase “This study provides valuable information for implementation and feasibility of FAST strategy in Nepal”.

7. PLOS authors have the option to publish the peer review history of their article (what does this mean?). If published, this will include your full peer review and any attached files.

Reviewer #3: **Yes: **Gidado Mustapha

Reviewer #4: **Yes: **Nino Maghradze

---

## [Author Response · Author response to Decision Letter 1]

26 Jun 2021

To, 

Associate Editor

PLOS ONE

Subject: Re-submission of the revision of manuscript PONE-D-20-20684R1

Respected Sir,

Thank you for your consideration of our manuscript entitled “Feasibility of Find cases Actively, Separate safely and Treat effectively (FAST) Strategy for Early Diagnosis of TB in Nepal: An Implementation Research” PONE-D-20-20684R1. We have reviewed the comments of the reviewers and have thoroughly revised our manuscript We are thankful to both reviewer 3 and reviewer 4 for the constructive comments and feedback on our manuscript. 

We found the comments, correction and suggestions received from reviewers are insightful and helped us in improving the manuscript. We have made attempt to fully address these comments and incorporate the feedback in the revised manuscript and believe our revised manuscript represents a significantly improved. Please find attached a point-by-point response to reviewers for your kind consideration. We hope that you find our responses satisfactory, and that the manuscript will be accepted for publication. 

Many thanks again. 

Sincerely,

Sagun Paudel, MPH

Response to Reviewers

Reviewer #4

Abstract

Point 1: line 24 OPD – change to full wording

Response: This has been updated with ‘Out-patient department’

Point 2: line 35 – rephrase “study identified that the current setting of implementation…”

Response: Updated as per suggestion.

Point 3: line 37-38 – rephrase – “…showed that hospital ownership is crucial/is important for mobilizing staff (not staffs)….”

Response: Rephrased as per suggestion.

Point 4: line 38 – remove “that”

Response: Removed.

Point 5: line 40 - cartridge not cartilage

Response: Corrected. 

Point 6: line 40 – manpower (not fair with women) = human resources

Response: Added as per suggestion.

Point 7: line 44 – rephrase “… effective function of the strategy”

Response: Rephrased as per suggestion.

Introduction

Point 8: line 50 – rephrase, it is not top ten it is among top ten, caused by single infectious agent, update WHO report date – 2017 to the newer version.

Response: Updated as per latest report. 

Point 9: line 52 - remove “which is steady in 2018” 

Response: Removed.

Point 10: line 57 – add “… among the leading causes of death in the country”

Response: Added as per suggestion.

Point 11: line 64 - rephrase – “Early case detection is important to interrupt infection transmission chain within healthcare facilities and community, in addition to reduce time of patients` suffering.”

Response: Rephrased as per suggestion.

Point 12: line 63 – NTC needs full wording

Response: Updated National Tuberculosis Control Center

Point 13: line 65 – change activity to process.

Response: Changed as per suggestion.

Point 14: line 68-70 - remove sentence starting with “Fast was initially…”

Response: Removed as per recommendation 

Point 15: line 71 – figure 1 – figure misses “No” in the upper right option of current cough and the options of two conditions such as: 1. Signs and symptoms – yes / chest x-ray –normal; 2. Signs and symptoms –no / chest x-ray abnormal; - what happens in such situations?

Response: Figure 1 has been updates and modified to make it simple. 

Point 16: line 73 – define DR- TB

Response: Added as Drug-resistant tuberculosis (DR-TB)

Point 17: line 77 – change early diagnosis to early detection

Response: Updated as per suggestion.

Point 18: line 82-88– rephrase to – “In order to suggest National Tuberculosis control program the areas which are crucial to be improved to strengthen the health facility-based diagnosis and prompt treatment of TB, we aimed to explore organizational, technical and financial factors needed to implement FAST activities in four hospitals.”

Response: Updated as per recommendation.

Materials and methods

Point 19: line 97 – define criteria for “purposively selected patients” – meaning they were selected based on…?

Response: Those patients, who attended the out-patient department with chest symptomatic features and screened for TB were selected for the interview purpose.

Point 20: line 92 – include FGD abbreviation with focus group discussion

Response: Updated as focus group discussion.

Point 21: Study setting – start paragraph with line 110 – “The FAST is a global fund-supported activity, where various non-governmental organizations are providing technical support to the …..” continue with line 106, “The FAST strategy was prepared by ….”

Response: Changed as per the suggestion. 

Point 22: line 115 change “The FAST approach” to “As FAST approach”

Response: Added as per suggestion.

Point 23: line 117 change “separated presumptive TB cases to collect sputum for GeneXpert and diagnosed patients were enrolled for the treatment”

Response: Corrected. 

Point 24: line 120 – difference or different?

Response: It is different, Corrected. 

Point 25: line 125 – remove part of the sentence starting from “because these hospitals…”

Response: Removed as per suggestion. 

Results

Point 26: line 188 remove “purposively”

Response: Removed.

Point 27: line 190 – define groups clearly, eg. i) implementation context, ii) identified factors for feasibility iii) barriers and enablers iv) way forward for the program implementation

Response: Result groups are defined as per suggestion.

Point 28: Table 1 – Patient who is 40 years old, is over 40 or below 40? Define age categories like ≥40 and <40 at least. Or 18-39 / 40 and up… Same goes to work experience length.

Response: Corrected as ≥ 40 and ≥ 17 in age and work experiences respectively.

Point 29: line 197 – respondent or respondents?

Response: It was respondents, corrected. 

Point 30: line 199-200 rephrase

Response: Rephrased as ‘‘National stakeholders described that FAST is implemented as a non-costed intervention as there is no financial implication for program implementation, where TB screening and treatment services are offered at different hospitals.’’

Point 31: line 219 – change “So, it is difficult” to “Therefore it is difficult to pick the patient, compared to the Teaching hospital and Achham hospital.”

Response: Revised as per suggestion.

Point 32: line 222-225 rephrase whole paragraph “…and physical space can be managed inside the hospitals.”

Response: Rephrased as ‘‘Participants explained that the implementation setting was not readily structured as per the requirements however they strongly argued that the hospital setting, and service operation process can be easily modified as per FAST guideline to implement the FAST approach and physical space can manage inside hospitals.’’

Point 33: line 228 –include “[sic]” at the end of every quote, indicating there might be grammatical problems and you are not interfering with corrections.

Response: Corrected. 

Point 34: line 231 – change “… increased in implementing” to “… increased in the selected hospitals.”

Response: Revised as per suggestion.

Point 35: line 232 remove chest and change “… are prioritized to be examined faster.”

Response: Changed accordingly. 

Point 36: line 242 – “as per the expectation at the initial stage…”

Response: Corrected.

Point 37: line 243 – “…results show that the TB…”

Response: Corrected per suggestion.

Point 38: line 246 – rephrase – “…at the hospitals was performed per protocol, however…”

Response: Updated as per suggestion.

Point 39: paragraph lines 250-259 – shorten too many repetitions. Needs just a few sentences.

Response: Rephrased as ‘‘As the FAST approach consists of series of activities in the hospital setting, our study identified that the actively screening, counseling, and sputum examination was conducted as per the framework of FAST strategy however separation of the patient from the queue and place them in the isolated or well-ventilated room was not followed as per the requirement of the program. It was found that the diagnosed cases were properly enrolled for DOTS. As per the FAST framework, the main problem was identified on the separation of patients. This study identified that the treatment of diagnosed TB cases was initiated timely and the patients were enrolled at DOTS therapy within hospitals.’’

Point 40: lines 272- 275 – unclear, please rephrase, avoid repetition of the specific words.

Response: Rephrased as ‘‘Participants repeatedly stressed that strong hospital ownership is a fundamental factor for staff mobilization, arrangement of the client flow system, registration desk, and waiting room. Similarly, hospital ownership is required for overall program implementation, monitoring, recording, reporting, and evaluation of the program. We identified that the hospital ownership was established in a community hospital and NGO-operated hospital to mobilize local resources for FAST whereas the government hospital didn’t realize it.’’

Point 41: line 282 – change stressed to defined

Response: Corrected.

Point 42: line 286 – rephrase “… limited space overcrowded with patients…”

Response: Corrected as per suggestion.

Point 43: line 311 – Respondent or respondents?

Response: Corrected as respondents.

Point 44: line 324 – explain the meaning of “birami sahayogi”.

Response: Birami sahayogi is the administrative staff mobilized to support patients for searching service rooms inside hospitals.

Point 45: line 327 – rephrase “… support staffs mobilized..” to “… support personnel mobilized…”

Response: Corrected as per suggestion.

Point 46: line 348 – remove “... so it was not necessary for FAST”, but would be better to rephrase the whole sentence.

Response: Removed as per suggestion and rephrased. 

Point 47: line 361- result waiting time is turn-around time or reporting time. Change “until the next day.”

Response: Changed.

Point 48: lines 362-363 – rephrase to “… the patients within a day, however…”

Response: Corrected.

Point 49: line 375 – add “the” – “… to the other institutions...”

Response: Corrected.

Point 50: line 386 – rephrase – “The results showed absence of uniformity on counseling….”

Response: Corrected.

Point 51: line 408 – rephrase voiced to pointed out. Remove “is required” at the end of the sentence.

Response: Corrected.

Point 52: line 425 – manpower = human resources / work force

Response: Corrected. 

Point 53: line 428 – full wording for IEC.

Response: Added. Information, education, and communication (IEC)

Point 54: line 438 - “… DOTS service …. was also noticed as the enabling factor”

Response: Corrected. DOTS service in the same hospital were also noticed as an enabling factor.

Point 55: line 441 - change stressed to highlighted

Response: Corrected. 

Point 56: line 453 – maybe easy adoptable instead of “doable and possible”

Response: 

Discussion – Overall, this section is too big. There are many unnecessary parts, please shorten considerably. Do not repeat results as they already are in the section above.

Response: Attempted to reduce the duplicate content from results. 

Point 57: line 464 “… support are fundamental factors…”

Response: Corrected.

Point 58: line 466 – “… prevent infection transmission in a hospital setting…”

Response: Corrected as per suggestion.

Point 59: line 472 – “… is necessary for the FAST strategy functioning, so the hospitals will be obliged to implement the process in healthcare facilities.”

Response: This has been revised. 

Point 60: line 485 – “… delay in the maintenance..,”

Response: Corrected. 

Point 61: line 502 – “…barrier to the proper…”

Response: Corrected.

Point 62: line 507 – “… and timely maintenance…”

Response: Revised.

Point 63: line 508 – “… were considered as barriers for…”

Response: Revised.

Point 63: lines 517-520 – remove and start “We could not access…”

Response: Removed and corrected.

Point 64: lines 529 -533 – remove and start “Our study showed that CFIR….”

Response: Removed as per suggestion.

Conclusion

Point 65: line 542 – rephrase “This study provides valuable information for implementation and feasibility of FAST strategy in Nepal”.

Response: Added as per suggestion.

Reviewer #3:

Point 1: It will be helpful to describe better the variables considered under organizational, technical & financial factors in the data collection and analysis section-line 123

Response: Variables considered for this study has been updated. Please refer 180 to 185. 

Point 2: Line 151-157 should all be part of the methodology

Response: It was updated as per the suggestion of previous reviewer. 

Point 3: The conclusion: line 529-532 especially linking the conclusion with early diagnosis is not supported by the study.

Response: Revised.

---

## [Decision Letter · Decision Letter 2]

21 Jul 2021

PONE-D-20-20684R2

Feasibility of Find cases Actively, Separate safely and Treat effectively (FAST) strategy for early diagnosis of TB in Nepal: An implementation research

PLOS ONE

Dear Dr. Paudel,

Thank you for submitting your manuscript to PLOS ONE. After careful consideration, we feel that it has merit but does not fully meet PLOS ONE’s publication criteria as it currently stands. Therefore, we invite you to submit a revised version of the manuscript that addresses the points raised during the review process.

Please submit your revised manuscript. If you will need significantly more time to complete your revisions, please reply to this message or contact the journal office at plosone@plos.org. Please include the following items when submitting your revised manuscript:

We look forward to receiving your revised manuscript.

Kind regards,

Frederick Quinn

Academic Editor

PLOS ONE

Journal Requirements:

Reviewers' comments:

Reviewer's Responses to Questions

**Comments to the Author**

1. If the authors have adequately addressed your comments raised in a previous round of review and you feel that this manuscript is now acceptable for publication, you may indicate that here to bypass the “Comments to the Author” section, enter your conflict of interest statement in the “Confidential to Editor” section, and submit your "Accept" recommendation.

Reviewer #3: All comments have been addressed

Reviewer #4: (No Response)

2. Is the manuscript technically sound, and do the data support the conclusions?

Reviewer #3: Yes

Reviewer #4: Yes

3. Has the statistical analysis been performed appropriately and rigorously? 

Reviewer #3: N/A

Reviewer #4: Yes

4. Have the authors made all data underlying the findings in their manuscript fully available?

Reviewer #3: Yes

Reviewer #4: Yes

5. Is the manuscript presented in an intelligible fashion and written in standard English?

Reviewer #3: No

Reviewer #4: No

6. Review Comments to the Author

Reviewer #3: As I early indicated, I have minimal competency to give a fair judgement on qualitative method. However, I studied both my feedback and that of other revivers and realized that the author have addressed all concerns.

However, this manuscript will need significant editing and use of scholarly tone, and in-text citation.

Reviewer #4: Comments to the author:

Thank you for addressing all the comments. I fully understand that the review process is long and labor consuming, however there are still minor corrections needed.

Please pay attention to the grammar and academic writing.

Line 25 – remove “settings” - change to “… TB transmission in hospitals”

Line 36 – “…current settings…”

Line 60-61 – re-phrase: “Tuberculosis is a major public health problem in Nepal, as it is ranked as the sixth among the leading causes of death in the country.”

Line 67 – add to deletion - “—tome of patient” (there is a repetition)

Line 93 – insert full stop after “...improve.”

Line 126- delete “…to the...” (There is a repetition)

Line 150 – correct – “Another study area/site…”

Line 243 – Remove – “So it is difficult” and start with “Therefore it is difficult”

Line 248 – Add comma after “requirements” change full stop to comma after “setting”

Line 253 – Correct “…physical space can be managed inside the hospitals”

Line 261- re-correct – “…as well as symptomatic patients are prioritized to examine faster.” Remove repetition.

Line 282 – Divide the sentence, use full stop after “strategy.” And start following sentence with “However separation…”

Line 409 – change waiting time to reporting time

Line 536 – change waiting time to reporting or turn-around time

Line 562 – remove “a” “were considered as barriers”

Lines 228, 237, 246, 256, 265, 268, 323, 339, 347, 380, 440, 453, 500 – at the end of each quote include [sic], e.g. “‘‘It is a very basic matter,…. Space management, a supportive laboratory staff, GeneXpert and one dedicated staff is enough to implement FAST. [sic] ’’ (Hospital focal person, community hospital, KII)

7. PLOS authors have the option to publish the peer review history of their article (what does this mean?). If published, this will include your full peer review and any attached files.

Reviewer #3: **Yes: **Gidado Mustapha

Reviewer #4: **Yes: **Nino Maghradze

---

## [Author Response · Author response to Decision Letter 2]

16 Sep 2021

Subject: Re-submission of the revision of manuscript PONE-D-20-20684R2

Respected Sir/Madam,

Thank you very much for this opportunity to submit a revised draft of manuscript entitled “Feasibility of Find cases Actively, Separate safely and Treat effectively (FAST) Strategy for Early Diagnosis of TB in Nepal: An Implementation Research” PONE-D-20-20684R2. We really appreciate the time and effort provided by Editors and reviewers and providing insightful feedback to improve our paper. We have reviewed the corrections and feedback received and and have thoroughly revised our manuscript.

We found the comments, correction and suggestions received from reviewers are insightful and helped us in improving the manuscript. Please find attached a point-by-point response to reviewers for your kind consideration. We hope that you find our responses satisfactory, and that the manuscript will be accepted for publication. 

Many thanks again. 

Sincerely,

Sagun Paudel, MPH

Response to Reviewers

Reviewer #4:

Line 25 – remove “settings” - change to “… TB transmission in hospitals”

Response: This has been updated.

Line 36 – “…current settings…”

Response: Changed. 

Line 60-61 – re-phrase: “Tuberculosis is a major public health problem in Nepal, as it is ranked as the sixth among the leading causes of death in the country.”

Response: Rephrased as per suggestion 

Line 67 – add to deletion - “—tome of patient” (there is a repetition)

Response: Deleted the repeated words ‘time of patient’

Line 93 – insert full stop after “...improve.”

Response: Added.

Line 126- delete “…to the...” (There is a repetition)

Response: Removed

Line 150 – correct – “Another study area/site…”

Response: Corrected. 

Line 243 – Remove – “So it is difficult” and start with “Therefore it is difficult”

Response: Removed. 

Line 248 – Add comma after “requirements” change full stop to comma after “setting”

Response: Changed.

Line 253 – Correct “…physical space can be managed inside the hospitals”

Response: Corrected. 

Line 261- re-correct – “…as well as symptomatic patients are prioritized to examine faster.” Remove repetition.

Response: Removed. 

Line 282 – Divide the sentence, use full stop after “strategy.” And start following sentence with “However separation…”

Response: Corrected 

Line 409 – change waiting time to reporting time

Response: Changed. 

Line 536 – change waiting time to reporting or turn-around time

Response: Changed to turn-around time. 

Line 562 – remove “a” “were considered as barriers”

Response: Removed

Lines 228, 237, 246, 256, 265, 268, 323, 339, 347, 380, 440, 453, 500 – at the end of each quote include [sic], e.g. “‘‘It is a very basic matter,…. Space management, a supportive laboratory staff, GeneXpert and one dedicated staff is enough to implement FAST. [sic] ’’ (Hospital focal person, community hospital, KII)

Response: Added

---

## [Decision Letter · Decision Letter 3]

8 Oct 2021

Feasibility of Find cases Actively, Separate safely and Treat effectively (FAST) strategy for early diagnosis of TB in Nepal: An implementation research

PONE-D-20-20684R3

Dear Dr. Paudel,

We’re pleased to inform you that your manuscript has been judged scientifically suitable for publication and will be formally accepted for publication once it meets all outstanding technical requirements.

Kind regards,

Frederick Quinn

Academic Editor

PLOS ONE

Additional Editor Comments (optional):

Reviewers' comments:

Reviewer's Responses to Questions

**Comments to the Author**

1. If the authors have adequately addressed your comments raised in a previous round of review and you feel that this manuscript is now acceptable for publication, you may indicate that here to bypass the “Comments to the Author” section, enter your conflict of interest statement in the “Confidential to Editor” section, and submit your "Accept" recommendation.

Reviewer #4: All comments have been addressed

2. Is the manuscript technically sound, and do the data support the conclusions?

Reviewer #4: Yes

3. Has the statistical analysis been performed appropriately and rigorously? 

Reviewer #4: (No Response)

4. Have the authors made all data underlying the findings in their manuscript fully available?

Reviewer #4: Yes

5. Is the manuscript presented in an intelligible fashion and written in standard English?

Reviewer #4: Yes

6. Review Comments to the Author

Reviewer #4: (No Response)

7. PLOS authors have the option to publish the peer review history of their article (what does this mean?). If published, this will include your full peer review and any attached files.

Reviewer #4: **Yes: **Nino Maghradze

---

## [Editor Report · Acceptance letter]

13 Oct 2021

PONE-D-20-20684R3 

Feasibility of Find cases Actively, Separate safely and Treat effectively (FAST) strategy for early diagnosis of TB in Nepal: An implementation research 

Dear Dr. Paudel:

I'm pleased to inform you that your manuscript has been deemed suitable for publication in PLOS ONE. Congratulations! Your manuscript is now with our production department. 

Kind regards, 

on behalf of

Dr. Frederick Quinn 

Academic Editor

PLOS ONE